# Quantifying vector diversion effects in zoonotic systems: A modelling framework for arbovirus transmission between reservoir and dead-end hosts

Emma L. Fairbanks[1,2]*, Matthew Baylis[3], Janet M. Daly[4], Michael J. Tildesley[1]

**1** The Zeeman Institute for Systems Biology and Infectious Disease Epidemiology Research, Mathematics Institute and School of Life Sciences, University of Warwick, Coventry, United Kingdom, **2** Department of Mathematics, University of Manchester, Manchester, United Kingdom, **3** Institute of Infection, Veterinary and Ecological Sciences, Faculty of Health and Life Sciences, University of Liverpool, Liverpool, United Kingdom, **4** One Virology - Wolfson Centre for Global Virus Research, School of Veterinary Medicine and Science, University of Nottingham, Loughborough, United Kingdom

* emma-louise.fairbanks@warwick.ac.uk

## Abstract

Vector-borne disease transmission involves complex interactions between vectors, reservoir hosts and dead-end hosts. We present a mathematical model for the vectorial capacity that incorporates multiple host types and their interactions, focusing specifically on West Nile virus transmission by *Culex pipiens* mosquitoes. Our model integrates climate-dependent parameters affecting vector biology with vector control interventions to predict transmission potential under various scenarios. We demonstrate how vector control interventions targeting one host type can significantly impact transmission dynamics across all host populations. By examining the effects of different vector control tool modes of action (repellency, preprandial killing, disarming and postprandial killing), we develop target product profiles that minimise unintended consequences of vector control. Notably, we identify the optimal intervention characteristics needed to prevent repellency on dead-end hosts from inadvertently increasing transmission among reservoir hosts. This research provides valuable insights for public health officials designing targeted vector control strategies and offers a flexible modelling framework that can be adapted to other vector-borne diseases with complex host dynamics.

## Author summary

Mosquitoes that spread diseases like West Nile virus don't just bite one type of animal—they feed on birds, humans, and other mammals. This creates a complex web of disease transmission that current prevention strategies often overlook. We developed a mathematical model to understand what happens when mosquito control methods target different types of hosts in this network.

**Data availability statement:** R code to simulate the model is available at https://github.com/emmafairbanks/EntoModels (DOI:10.5281/zenodo.17602796).

**Funding:** The authors were supported by the Horserace Betting Levy Board (vet/prj/809) to MJT. MJT is funded on a joint BBSRC/EEID grant (BB/T004312/1). The funders had no role in study design, data analysis, decision to publish or preparation of the manuscript.

**Competing interests:** The authors have declared that no competing interests exist.

Our research reveals a surprising and concerning finding: when people use repellents to protect themselves from mosquitoes, those mosquitoes don't simply disappear—they redirect to birds instead. Since birds are the main animals that can spread West Nile virus to other mosquitoes, this redirection can actually increase disease transmission in the bird population by up to 23%. More infected birds ultimately means more infected mosquitoes and higher risk for humans. However, we also identified a solution. We found that if repellent products could kill just 2% of the mosquitoes they encounter before those mosquitoes find alternative hosts, this would eliminate the harmful redirection effect. Our work provides specific guidelines for developing better mosquito control products and helps public health officials understand the broader consequences of different intervention strategies. This framework can be applied to other mosquito-borne diseases beyond West Nile virus.

## 1 Introduction

For vector-borne diseases, the basic reproductive number (the expected number of cases to arise from a single infectious case in a susceptible population) can be expressed as the product of the vectorial capacity (a measure of the potential of a vector population to transmit a pathogen) and duration of the host's infectious period [17]. Many mathematical models for predicting the risk of disease transmission consider the effects of climate and vector control on the number of vectors [1–15] as a proxy for vectorial capacity. However, vectorial capacity, a measure of the potential of a vector population to transmit a pathogen, is defined as the total number of potentially infectious bites that would arise from all vectors biting a single infectious host on a single day [16]. Therefore, vectorial capacity should not only consider the number of vectors, but also their ability to transmit a pathogen, and is therefore the product of both the number of vectors and the relative vector capacity (rVC), defined as the total number of potentially infectious bites that would arise from a single vector biting a single host on a single day (Fig 1).

The basic reproductive number, the expected number of cases to arise from a single infectious case in a susceptible population, can be expressed as the product of the vectorial capacity and duration of the host's infectious period [17].

We previously modelled the rVC for viruses transmitted by *Culicoides* [18], considering how climatic variables affect the rate at which vectors feed, expected lifespan of vectors and the extrinsic incubation period (EIP), defined as the time required for a vector to become infectious after consuming an infected blood meal. The model also considers the effects of vector control interventions at different stages during the mosquito feeding cycle. Additionally, the model considers host selection, which is influenced by the availability of different host types and the vector's preferences for these hosts. However, the model only considers how the vectors interact with one type of susceptible host. In this study, we expand this model to address this limitation

PLOS Computational Biology

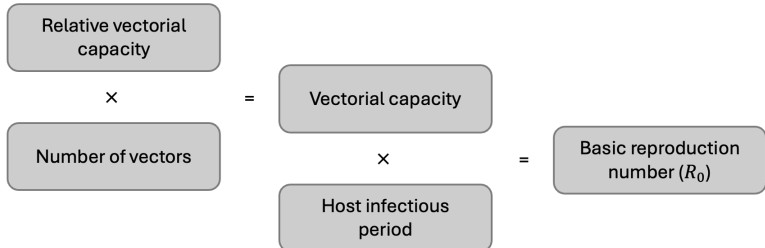

**Fig 1**. **Schematic describing the relationship between the number of vectors, relative vectorial capacity, vectorial capacity, host infectious period and basic reproduction number.**

and consider how the vectors interact with multiple susceptible host species, and how a change in behaviour related to one may influence transmission dynamics to others.

However, there are some limitations to this model. For example, the model only considers one host species. While this may be informative in localised populations with only one competent host present, it is not possible to consider interactions between host species. In some cases, a disease may be mainly transmitted to a host which shows less severe clinical signs than other hosts or dead-end hosts (hosts which can be infected but cannot infect). Hosts which can both be infected and transmit a pathogen are often referred to as reservoir hosts. These hosts can amplify disease transmission, whereas infection of a dead-end host has no downstream effects of transmission. In these cases it is important to consider the interactions between hosts.

Vector control interventions targeting one host type can significantly impact transmission dynamics across all host populations. When control measures such as repellents or insecticides are applied to one host type, mosquito vectors may be diverted to alternative hosts, potentially altering disease transmission patterns.

In this study, we extend this model to consider the rVC for transmission of pathogens between host species. Here, we include reservoir hosts and dead-end hosts. As an example, we model West Nile virus (WNV) transmission by *Culex pipiens*, where interactions between reservoir hosts (birds) and dead-end hosts (humans and other mammals) determine overall transmission intensity.

## 2 Methods

### 2.1 Model development

The model for simulating the dynamics in a single host species is described in full detail in [18]. Here, the rVC on day $t$ was calculated as the probability the vector feeds on the host of interest on day $t$ and survives until the end of that day multiplied by the probability it infects another of the same host type every day after, given it survives until that day. The model considered the climatic dependence of the rates of gonotrophic cycle completion, vector mortality and pathogen EIP completion. It can either be simulated using daily climatic data or, alternatively, for a constant temperature or set durations of gonotrophic cycle completion, vector lifespan and EIP completion.

In this study, we extend this model to account for the interactions between vectors and different host types. This multiple-host model is illustrated in Fig 2 and described in detail in S1 Text. It can be utilised to calculate the transmission potential from each reservoir host species to each other host species, including dead-end hosts. For interpretability, we define the basic reproduction number ($R_0$) as the product of the rVC, the infectious period and the expected number of vectors per host [17,18].

Within the modelling framework, vectors that encounter hosts can experience several endpoints: they might successfully feed, be killed before (preprandial mortality), become disarmed (prolonged blood feeding inhibition) or abandon the

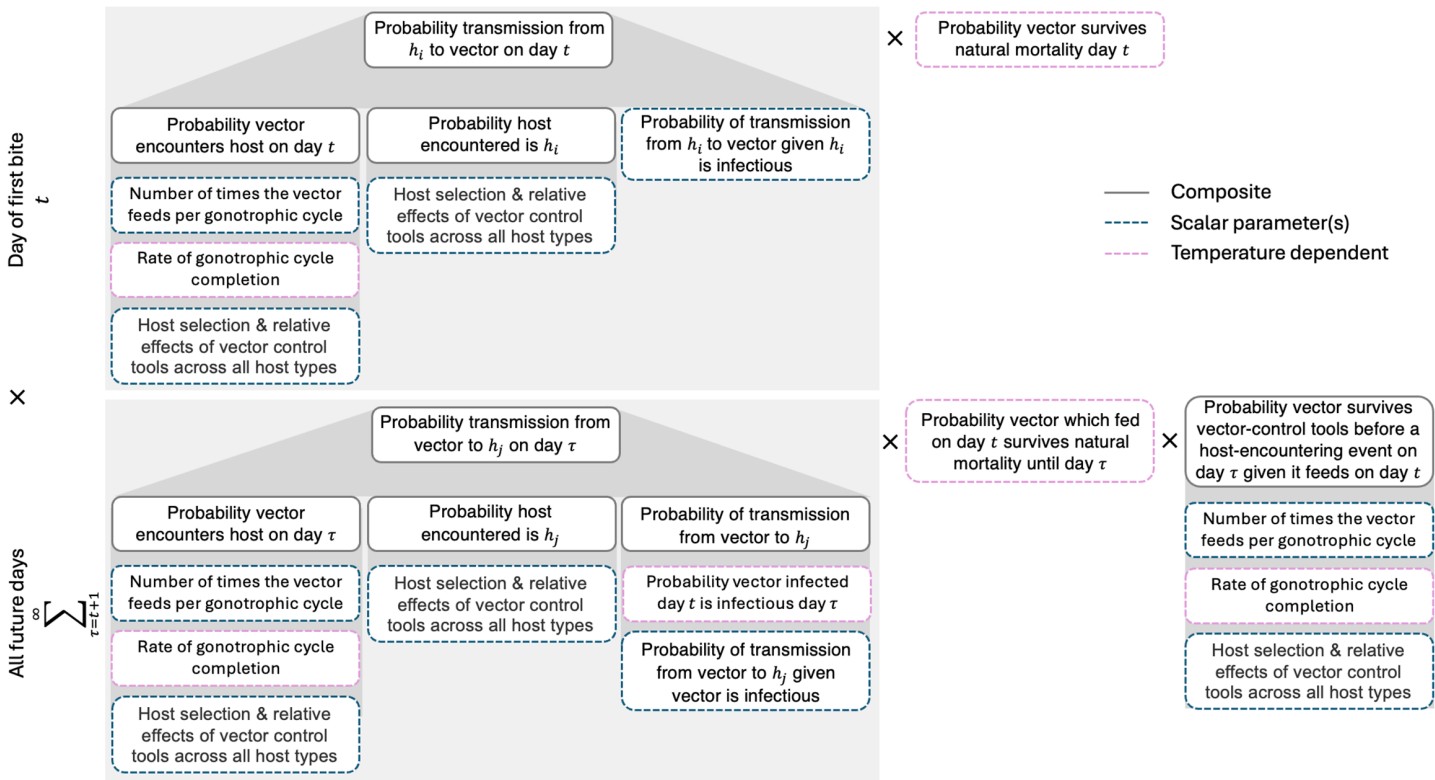

**Fig 2**. Schematic representation of relative vectorial capacity (rVC) calculation showing the probability of transmission from a reservoir host ($h_i$) through a vector to another host ($h_j$). The top panel shows the probability of host to vector transmission on day $t$, and the bottom panel shows the probability of vector to host transmission on all future days $\tau > t$. The rVC is the product (×) of these probabilities. Boxes show how composite parameters (solid lines) are calculated from scalar parameters (dashed lines) and temperature-dependent parameters (purple dashed lines), accounting for vector control interventions across multiple host types.

current host to search for another. Vectors that succeed in feeding may still be killed after the blood meal (postprandial mortality).

The model simulations and subsequent data processing were conducted in R, with visualisations generated using the ggplot2 package [19] within the RStudio integrated development environment [20].

## 2.2 Model application

We simulate the model for West Nile virus transmission by *Cx. pipiens*. The model parameters were set following a comprehensive literature review. The rate of gonotrophic cycle completion and vector life-span per day were parameterised by [21] using data from four [22–25] and three [24,26,27] *Cx. pipiens* studies, respectively. For the EIP, we use the model from [28], which was parameterised using Bayesian methods and data from 2145 mosquitoes. [28] found that the gamma and Weibull distributions were comparable in their ability to fit the EIP data. Here, we favour the gamma distribution since it has a property of being able to be broken up into multiple exponential distributions. This allows for consideration of daily fluctuations in temperature.

When estimating the number of vectors per host required for $R_0 > 1$, we consider avian reservoir hosts with infectious periods of 7 and 10 days [29]. An infectious period of 7 days may be applicable to geese [30], passerines [31] and owls [32], and 10 days is applicable to raptors [33] and turkeys [34].

**2.2.1 Comparing the transmission potential between host types.** [35] reviewed blood meal sources from 26,857 *Culex* species, including 2,062 *Cx. pipiens* in palearctic regions. They found that 68.3% percent had fed on avian hosts, and 14.1% and 17.2% fed on human and non-human mammal hosts, respectively. We consider the rVC from birds (the reservoir hosts) to human and other mammal hosts (assuming these are a dead-end hosts). As not all mammals are hosts of WNV this is likely to represent an upper bound. We calculate the rVC for temperatures between 10 and 32 °C using these fixed blood selection values, however these are likely to vary between settings [36].

The probability of transmission from vector to host is assumed to be the same for all hosts (0.74) [37]. We also assume that vectors bite only once per feeding cycle and that there are no vector control tools present.

**2.2.2 Simulation at a larger spatial scale.** We then simulate the model considering daily temperature time series from the UK to produce maps showing the patterns in transmission potential. Temperature for 2019–2023 was extracted from the HadUK-Grid gridded average climate observations for 5 km grid squares of the UK [38]. For each year, we calculate the rVC each day. To summarise the results, we pool estimates for each month across days and year to calculate the monthly minimum, mean and maximum. The UK represents an ideal case study for demonstrating applicability of the model, as it encompasses considerable climatic variation across a relatively compact geographical area, allowing for examination of diverse transmission dynamics.

**2.2.3 Considering the effect of vector-control tools on different host types.** When considering how vector control affects the transmission potential, we will consider two host types; a reservoir (birds) and another dead-end host. We vary the percentage of vectors feeding on the reservoir host and assume the rest feed on a dead-end host. We simulate the model for settings where, in the absence of vector control, the vector would select a reservoir host 20%, 50% or 80% of the time to capture a range of demographic settings.

Initially, we will consider the effects of a tool which only repels mosquitoes, therefore they continue host-seeking and may bite another host of any type. We vary the repellent effect assuming a 0–100% reduction in the rate of blood feeding. The coverage is also varied from 0-100% of the target host type having the tool. We assume that if a host type has access to a tool they always use it when exposed to mosquitoes. When analysing the effects of vector-control we fix the temperature at the optimal value for transmission, as calculated in Sect 2.2.1.

Finally, we investigate how other vector control characteristics (modes of action) can counteract the potential negative effects of repellents. When repellents are applied to dead-end hosts, they may inadvertently increase transmission by redirecting mosquitoes towards reservoir hosts. For each percentage of the reduction in the rate of biting investigated (0–100%), we therefore calculate the percentage of the repelled mosquitoes which would need to follow each alternative mode of action (preprandial killing, disarming, or postprandial killing) to mitigate this increased transmission risk.

# 3 Results

## 3.1 Model development

Our study extends a single-host mathematical model of the rVC that integrates climatic factors and vector control interventions, introducing interactions between vectors and multiple types of host. This multiple-host model enables consideration of secondary and dead-end hosts in the analysis allowing for consideration of how control tools on one host type affect transmission potential to other host types, as demonstrated in the following results.

## 3.2 Comparing the transmission potential to host types

As expected, the magnitude of the rVC for each type of host corresponds to the likelihood a vector selects the host, with hosts fed upon more often receiving a larger portion of the disease burden (Fig 3). The maximum rVC is when the temperature is 24.5 °C. This corresponds to a rVC of 0.061, 0.013 and 0.015 for birds, humans and other mammals, respectively.

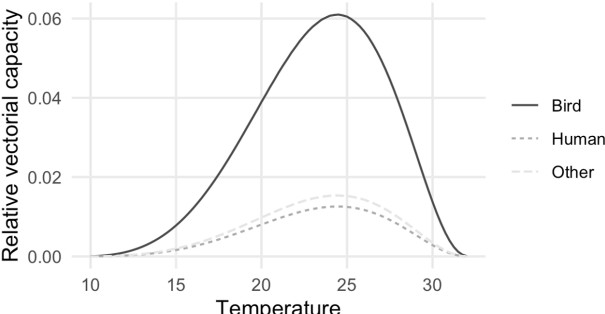

**Fig 3**. **The relative vectorial capacity from birds (the reservoir host) to each host type at constant temperatures.**

For a bird species with an infectious period of 7 days, the vectors per host required for $R_0 > 1$ in birds, humans and other hosts would be 2.34, 10.99 and 9.52, respectively. For bird species with an infectious period of 10 days the required number of vectors per host would be 1.64, 7.69 and 6.67 for birds, humans and other hosts, respectively.

### 3.3 Large scale simulation

Using daily mean temperature data, we produced maps of the mean monthly rVC (Fig 4). This allows for comparisons both spatially and temporally. Further details regarding monthly rVC values and corresponding vector population thresholds can be found in S1 Table.

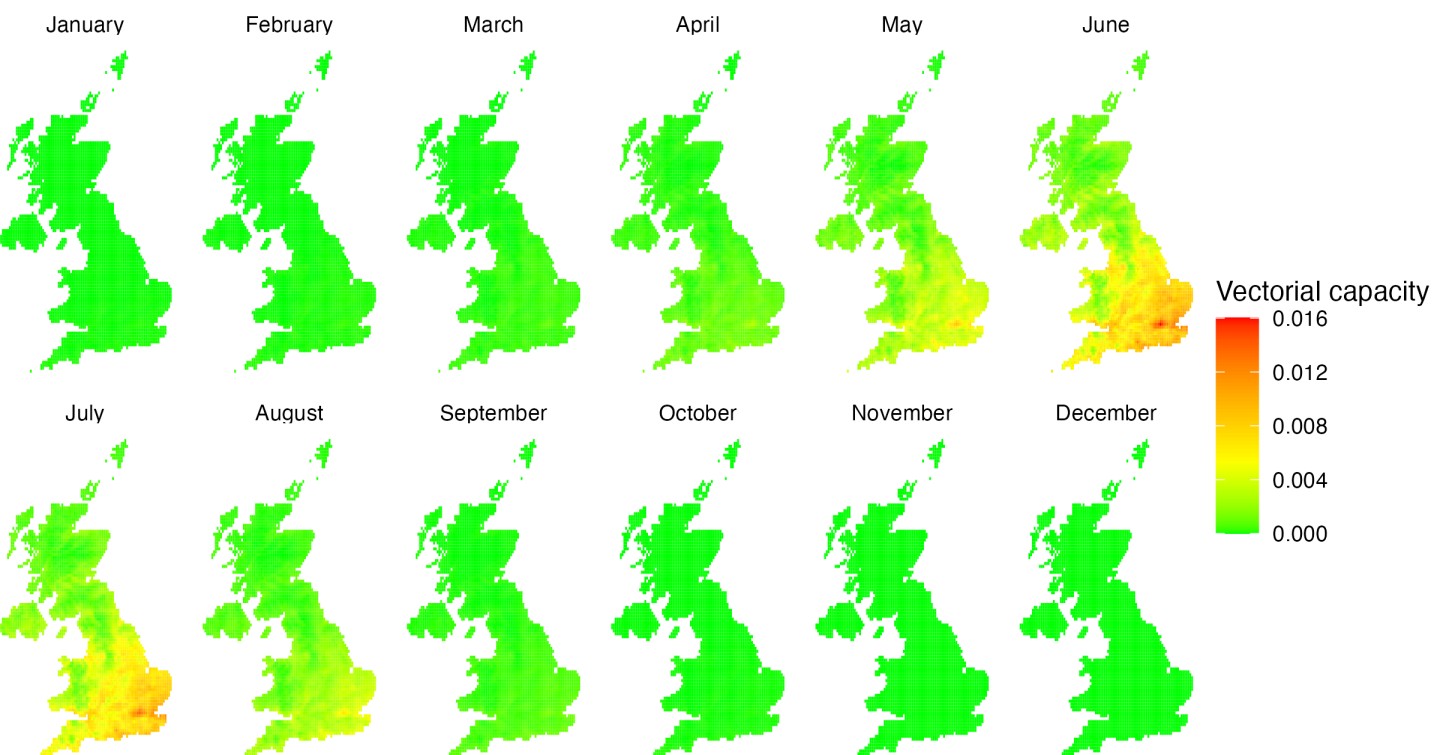

**Fig 4**. **Monthly mean predicted relative vectorial capacity for birds (the reservoir host) calculated for mean daily temperature estimates for 2019–2023 [38].**

The modelling outputs reveal considerable spatial heterogeneity. We observe that the rVC in Scotland and Northern Ireland is relatively low when compared with Wales and England, with England having the largest predicted rVC on average. The model predicts the largest rVC in south-east England, however, similar values are observed across the south of England, in South Wales and as far north as Merseyside and parts of Yorkshire.

The estimates demonstrate pronounced seasonal variations in rVC with significant implications for disease transmission dynamics. June exhibits the peak rVC (0.016), situated within a broader high-transmission window extending from May to August. During this summer period, the required vector population threshold for disease establishment ($R_0 > 1$) remains biologically plausible, ranging from approximately 6 to 15 vectors per host for an avian reservoir host with a 10-day infectious period.

In contrast, the winter months (November to March) show markedly reduced transmission potential, with numerous locations experiencing a rVC of zero. The maximum rVC in December is approximately 1500 times lower than the June peak, corresponding to an extraordinarily high vector population threshold (9371.54 vectors per host) that renders sustained transmission biologically unlikely given typical winter vector population densities. April and September–October emerge as critical transition periods, characterised by rapidly changing rVC as environmental conditions shift.

### 3.4 Considering the effect of vector-control tools on different host types

For interpretation, it is important to consider that, given a fixed number of vectors per host within a population, we would expect the vectorial capacity to increase or decrease proportionally with the rVC. This assumes that the size of the vector population is limited by the carrying capacity, which is a common assumption when modelling the impact of interventions on vectorial capacity [39–41].

Fig 5 shows that while repelling vectors from a reservoir host always reduces the rVC of the reservoir and dead-end hosts, repelling vectors from dead-end hosts can increase the rVC of reservoir hosts. As coverage of the tool and repellency increase we see a larger decrease in rVC for both host types when the tool is given to a reservoir host or in dead-end hosts when the tool is given to a dead-end host, and a larger increase in the rVC in a reservoir host when the tool is given to a dead-end host.

When targeting dead-end hosts, the reduction in the rVC of dead-end hosts tells us that each mosquito that feeds on an infectious reservoir is less likely to cause an infection in the dead-end host. However, the unintentional increase in vectorial capacity in reservoir hosts suggests that there will be more infected reservoir hosts, and therefore potentially more vectors becoming infected. In our scenario, where the model is parameterised to describe WNV transmission by *Cx. pipiens*, when dead-end hosts are protected with a purely repellent tool (no killing or disarming effects) at 100% coverage and 100% reduction in biting rate, vectorial capacity among reservoir hosts increases by up to 23.4%.

### 3.5 Target product profiles for vector-control tools

For a given level of repellency for a tool used on a dead-end host, Fig 6 shows the estimated magnitude of the other modes of action required to avoid the increase in vectorial capacity for reservoir hosts. Tools which preprandially kill at least 2% of *Cx. pipiens* of the repelled vectors counteract these unintended negative effects for all levels of repellency. However, for tools which repel almost all (>96%) mosquitoes feeding on dead-end hosts, postprandially killing or disarming cannot counteract all of the negative effects of the repellency.

Postprandial mortality is more effective than disarming, however its effects rapidly decrease when over 80% of vectors are repelled. The percentage of mosquitoes which need to be postprandially killed, given 80% or 90% of mosquitoes are repelled, is 8% or 18%, further increasing to 38% when 95% of mosquitoes are repelled. This is because the vectors must feed for postprandial killing to be effective – the fewer vectors that feed, of those that do feed a higher percentage need to be killed.

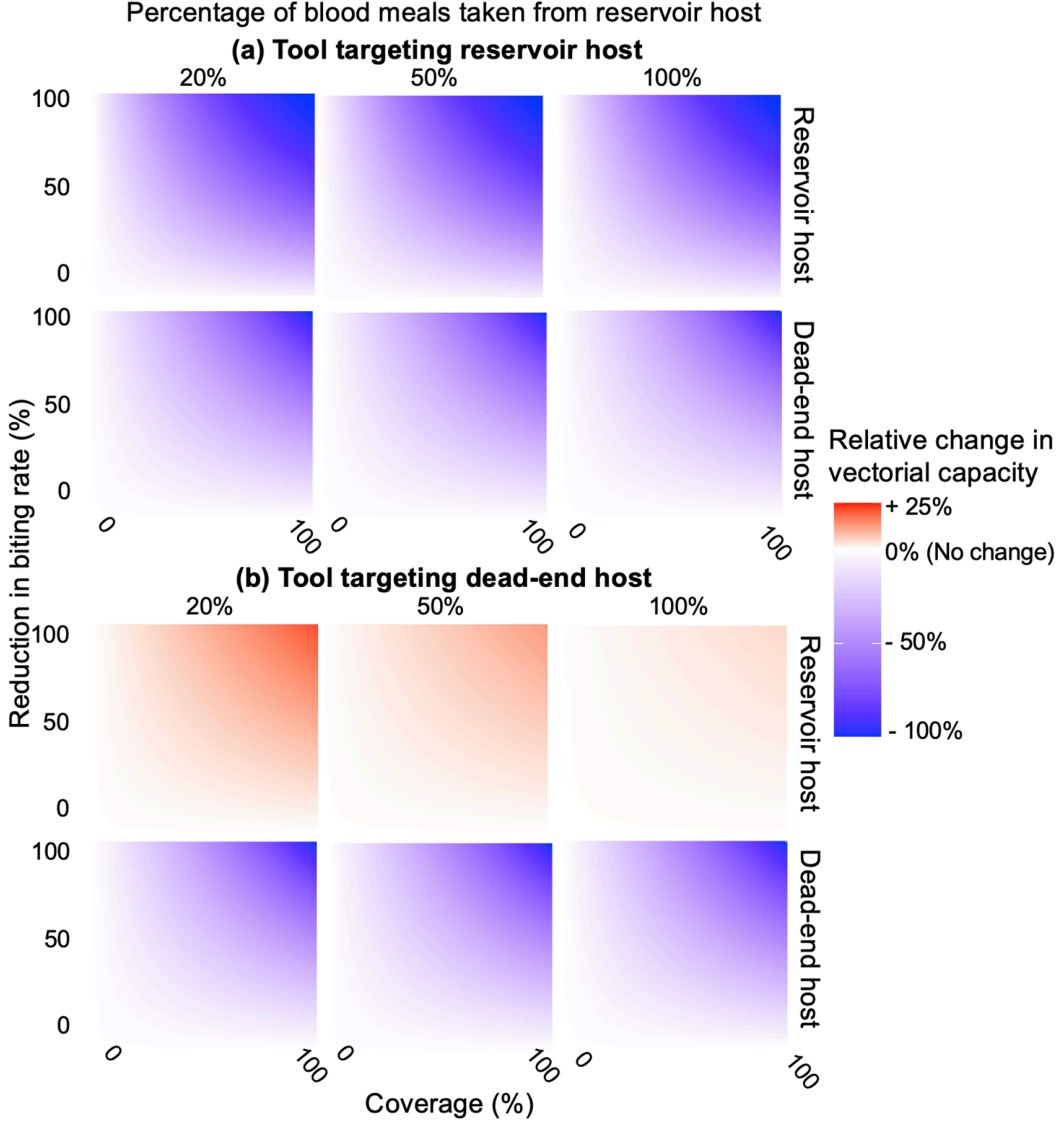

**Fig 5**. **The estimated relative change in vectorial capacity for the reservoir and dead-end host when a tool is applied to a (a) reservoir or (b) dead-end host for tools which reduce the rate of biting at different population coverage levels.** Columns show three scenarios representing the percentage of blood meals taken from reservoir hosts (birds): 20%, 50%, and 100%.

## 4 Discussion

To our knowledge, this is the first model that quantifies the rVC between multiple host types. Understanding these complex interactions is crucial for developing effective vector control strategies, as interventions focused solely on protecting dead-end hosts could inadvertently increase transmission among reservoir hosts, potentially elevating overall disease risk. By modelling these relationships, we can identify optimal intervention characteristics that minimise unintended consequences of vector control and maximise public health benefits.

The optimal temperature for WNV transmission was estimated to be 24.5°C, consistent with previous estimates [21,42], representing the thermal optimum where rVC peaks due to accelerated mosquito development, shortened

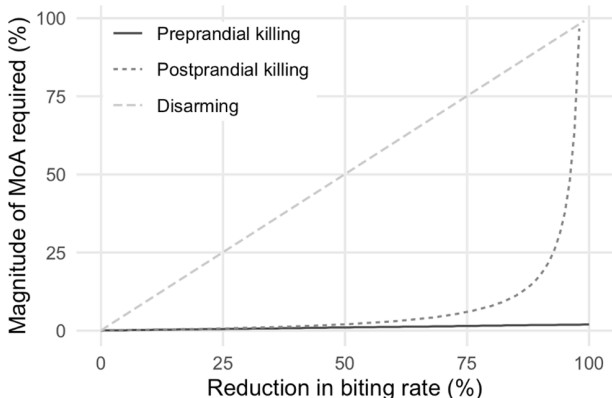

**Fig 6**. **Magnitude of each mode of action (MoA) required to counteract the increased vectorial capacity in reservoir hosts when a tool repells hosts from dead-end hosts.**

gonotrophic cycle, and reduced EIP while maintaining sufficient vector longevity. This optimal temperature falls within the summer temperature range of many temperate regions, including the UK, explaining observed seasonal patterns and suggesting that early warning systems and control strategies would benefit from targeted implementation during the high-transmission windows. Our model predicts higher rVC in urban areas, particularly London, the urban heat island effect creates temperatures 1–3 °C higher than rural environments [43,44], potentially extending the transmission season in metropolitan areas. However, disease transmission risk depends on both rVC and vector abundance, which is influenced by factors beyond temperature including rainfall , habitat availability and urban landscape heterogeneity [45], meaning high rVC does not necessarily translate directly to high transmission risk.

Host selection of vectors depends on host preference and host availability [18], therefore is likely to vary across settings. Experimental studies have demonstrated that *Cx. pipiens* prefer some avian species over others [46]. However, these preferences are modulated by environmental conditions and host availability in field settings, leading to spatial and temporal variation in feeding patterns [47]. Such heterogeneity in host selection across different landscapes significantly impacts WNV transmission potential. While our model accounts for heterogeneous vector preferences between host types through the blood index, it assumes spatial homogeneity within each host type, with all individuals of a given species equally likely to be bitten. Spatial clustering of hosts or individual variation in attractiveness could further modulate transmission patterns.

This modelling framework is flexible for application to other vectors and pathogens. Due to a reduction in wildlife habitats, we are observing an increase in the risk of human exposure to emerging and existing zoonotic pathogens [48]. An increase in transmission between wildlife reservoirs and human hosts may also have implications for disease elimination. Currently, numbers of human zoonotic malaria cases are increasing globally and there are currently no control measures that target wildlife reservoirs [48,49]. Parameterising the model for these pathogens could lead to further exploration of these dynamics.

Many health agencies suggest using repellent to reduce the risk of vector-borne disease [50–52]]. Our study shows that for zoonotic diseases this may not always be the case. More investigation is needed into the effects of increasing vectorial capacity on disease transmission. The modes of action of vector control tools change according to the resistance level of vectors and insecticide dose [39,40]. Insecticide resistance has been reported in *Cx. pipiens* in Europe [53] and globally [54]. Therefore, more data on the responses of local vector populations to insecticides should be gathered to ensure robust evaluation within different settings. This modelling framework can be used as an initial indicator of whether these tools are suitable for use on dead-end hosts, and suggest target product profiles of additional tools for use.

Repellents that also kill or disarm mosquitoes are more likely to avoid unintentional increases in vectorial capacity in reservoir hosts, with preprandial killing being the most effective. This study provides the first quantitative estimates of the magnitude of unique modes of action of tools required to reduce disease transmission. These quantitative thresholds offer valuable targets for product developers and public health officials designing next-generation vector control strategies. In our WNV example, if 2% of the reduction in biting is due to preprandial killing, vectorial capacity will be reduced in all hosts. However, this may vary across different settings and vector populations [41].

Our mathematical modelling presented here advances the understanding of host-vector-pathogen interactions. This work demonstrates the critical importance of considering entire transmission systems when implementing vector control, as interventions targeting one host population can significantly influence transmission throughout the ecological network. This underscores the need for integrated approaches to vector control that consider both personal protection and community-level impacts. The model is a valuable tool for assessing transmission risk and optimising control interventions in the face of emerging zoonotic threats and changing climate conditions.

## Supporting information

**S1 Text. Includes a description of the model and parameterisation for the West Nile virus application.**
(PDF)

**S1 Table. Gives the mean, minimum and maximum predicted monthly rVC values and corresponding vector population thresholds.**
(TIFF)

## Author contributions

**Conceptualization:** Emma L. Fairbanks, Matthew Baylis, Janet M. Daly, Michael J. Tildesley.

**Data curation:** Emma L. Fairbanks.

**Formal analysis:** Emma L. Fairbanks.

**Funding acquisition:** Emma L. Fairbanks, Janet M. Daly, Michael J. Tildesley.

**Investigation:** Emma L. Fairbanks.

**Methodology:** Emma L. Fairbanks, Michael J. Tildesley.

**Project administration:** Emma L. Fairbanks.

**Software:** Emma L. Fairbanks.

**Validation:** Emma L. Fairbanks.

**Visualization:** Emma L. Fairbanks.

**Writing – original draft:** Emma L. Fairbanks.

**Writing – review & editing:** Emma L. Fairbanks, Matthew Baylis, Janet M. Daly, Michael J. Tildesley.

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
