## [Decision Letter · Decision Letter 0]

28 Oct 2025

PCOMPBIOL-D-25-01479

Quantifying vector diversion effects in zoonotic systems: A modelling framework for arbovirus transmission between reservoir and dead-end hosts

PLOS Computational Biology

Dear Dr. Fairbanks,

Thank you for submitting your manuscript to PLOS Computational Biology. After careful consideration, we feel that it has merit but does not fully meet PLOS Computational Biology's publication criteria as it currently stands. Therefore, we invite you to submit a revised version of the manuscript that addresses the points raised during the review process.

Please submit your revised manuscript within 60 days Dec 28 2025 11:59PM. If you will need more time than this to complete your revisions, please reply to this message or contact the journal office at ploscompbiol@plos.org. Please include the following items when submitting your revised manuscript:

We look forward to receiving your revised manuscript.

Kind regards,

Jennifer A. Flegg

Section Editor

PLOS Computational Biology

Jennifer Flegg

Section Editor

PLOS Computational Biology

**Journal Requirements:**

Potential Copyright Issues:

- Figure 2. Please (a) provide a direct link to the base layer of the map (i.e., the country or region border shape) and ensure this is also included in the figure legend; and (b) provide a link to the terms of use / license information for the base layer image or shapefile. We cannot publish proprietary or copyrighted maps (e.g. Google Maps, Mapquest) and the terms of use for your map base layer must be compatible with our CC BY 4.0 license.

5) When completing the data availability statement of the submission form, you indicated that you will make your data available on acceptance. We strongly recommend all authors decide on a data sharing plan before acceptance, as the process can be lengthy and hold up publication timelines. Please note that, though access restrictions are acceptable now, your entire data will need to be made freely accessible if your manuscript is accepted for publication. This policy applies to all data except where public deposition would breach compliance with the protocol approved by your research ethics board. If you are unable to adhere to our open data policy, please kindly revise your statement to explain your reasoning and we will seek the editor's input on an exemption. Please be assured that, once you have provided your new statement, the assessment of your exemption will not hold up the peer review process.

**Reviewers' comments:**

Reviewer's Responses to Questions

**Comments to the Authors:**

Reviewer #1: Article Review – Quantifying vector diversion effects in zoonotic systems: A modelling framework for arbovirus transmission between reservoir and dead-end hosts (Fairbanks, Baylis, Daly, and Tildesley)

Abstract

•Great! Clear, concise and well-written.

Author summary

•Nice. Clear explanation of problem and solution you have found.

Introduction

•Definition of vectorial capacity and relation to basic reproductive number could be made more clear. Can you provide a simple example, equation, or illustration? Or perhaps start from your explanation of the basic reproductive number (the last sentence of the second paragraph), as this is a familiar concept that can be broken down into the parts you define.

•For consistency, it looks like you’re missing a line skip between the first and second paragraph.

•At the start of paragraph three, please clarify that you are referring to the one-host limitation from you previous model, and that addressing this limitation is a motivation for your current study.

Methods

Model development

•Please find a consistent way to distinguish the two models throughout. For example, “baseline” vs. “extended”, or “single-host” vs. “between-host”/“multiple host”, or something along those lines.

•It might be helpful to provide the rVC equation in paragraph one.

Model application

•Nice, I appreciated your explanation of why a gamma distribution was chosen.

•Please be consistent with use of commas, if you write 26,000, then 2000 should be 2,000.

•Great, this section explains the study and assumptions very well.

Results

•Well presented, Figure 2 is especially nice.

•I suggest using a different line style or color for “Disarming” in Figure 4, it’s very difficult to distinguish from “Postprandial killing.” Also, consider adding “due to repellant” to the x-axis label.

Discussion

•Include a citation for the “urban heat island effect” in paragraph three.

•I might suggest combining paragraphs two and three, there is a lot of information here about temperature that could be condensed. It detracts slightly from your main point and star contributions described in the last three paragraphs.

•Missing citation in paragraph four: [?]

Supplementary Information

•Missing reference after Equation (2)

* Be sure to make your code available on GitHub

Reviewer #2: The article is well written and logically structured. The analysis appears to represent a novel extension to existing methods. I would like to see several issues addressed before the paper is further considered for publication.

Major concerns:

• Lack of uncertainty analysis and reporting. All key results are reported as point estimates without any consideration of potential uncertainty.

• Insufficient methodological detail is provided in the main text. I suggest content is shifted from the supp material to the main text.

• Limitations of the analysis are not described in the discussion section and very little reference is made to other similar work.

Please provide line numbers on future submissions to assist with the review process.

Other comments:

2.1 Model development

Insufficient methodological details have been provided on this section. At minimum, I suggest that the authors shift the model equations from the supp material into the main text.

2.2. Model application

“vector life-span p day” should be “per day”?

“there we set ph->v = 1”. Define this parameter.

“…parameterised using Bayesian methods and data from 2145 mosquitoes”. Why state the methods for this study and not the others?

“This parameterisation includes the probability of transmission from host to vector and therefore we set ρh→v = 1 for reservoir hosts.” I don’t follow this logic. Why is the probability equal to 1?

“Vollans et al. [28] found that the gamma and Weibull distributions performed similarly”. Just confirming that you mean these distributions were equally good fits for the EIP?

“We calculate the rVC for temperatures between 10 and 32 degrees”. What temperature value? Daily minimum? Daily maximum? Daily mean? Monthly mean?

2.2.2

This is the first mention of the model being applied in the UK. Why is the model applied in the UK? No local cases of WNV have ever been reported in the UK (although it has been found in mosquitoes). I suggest that you add this context to the introduction to motivate your analysis. What are the possible reservoir host species in this setting?

What do you mean by “simulate” in this section?

3 Results

3.1 Model development – consider deleting this section. It should be covered in the methods.

3.2

Figure 1 – the short dash and long dash lines are very similar. Please use different indicators.

3.4

“In our scenario, where the model is parameterised to describe WNV transmission by Cx. pipiens, we see an increase in vectorial capacity of up to 23.4%.” Given what assumptions about the intervention? These must also be stated.

Figure 3 caption. “When a tool is used by…” Should this be when a tool is “applied to”?

“Percentage of blood meals taken from reservoir host” – this not described in the Figure 3 caption – please add.

Figure 4: Again, short and long dash lines are too similar. Please change.

4. Discussion

“Independently” – what do you mean by this?

“in line with previous estimates”. Please elaborate here. What did these studies find? What was their approach? How it is similar/different from your approach?

“Many health agencies suggest using repellent to reduce the risk of vector-borne disease [48–50]]. This work shows that for zoonotic diseases this may not always be the case.” The use of “this” in sentence two is confusing. Consider “our study” and re-state the finding that you refer to. Also, I would be very careful here. While you show that rVC can increase when repellent is used by human dead-end hosts, the individuals using repellent are still protected from disease exposure. Also worth considering the important limitations of your work here. You have only considered two hosts and the effect of repellent use in one of those hosts. What about the impact of other wildlife dead-hosts? Bites could be diverted to those hosts instead of the reservoir hosts (i.e. zooprophylaxis). Please discuss.

“This modelling framework is flexible for application to other vectors and pathogens.” Can you say a little more about how your model framework would be applied to the two example systems listed?

“This modelling framework can be used to assess whether these tools are suitable

for use on dead-end hosts, and suggest target product profiles of additional tools for use.” I don’t follow this point, can you say more here?

Supp methods

?? – some citations are missing

Beta is not described.

Equation 3 and Table S2. How are day s and day t different? I got a little confused here.

“…considered the climatic dependence of the rates of gonotrophic cycle completion, vector mortality and pathogen extrinsic incubation period (EIP) completion”. You also consider these dependencies in your work, please described them. Or at least point to section 1.2.1?

What “temp” was used for parameterisation in equations 11, 12, and 13?

**Have the authors made all data and (if applicable) computational code underlying the findings in their manuscript fully available?**

Reviewer #1: Yes

Reviewer #2: Yes

PLOS authors have the option to publish the peer review history of their article (what does this mean?). If published, this will include your full peer review and any attached files.

Reviewer #1: No

Reviewer #2: No

**Figure resubmission:**
---

## [Decision Letter · Decision Letter 1]

1 Dec 2025

Dear Dr Fairbanks,

We are pleased to inform you that your manuscript 'Quantifying vector diversion effects in zoonotic systems: A modelling framework for arbovirus transmission between reservoir and dead-end hosts' has been provisionally accepted for publication in PLOS Computational Biology.

Best regards,

Jennifer A. Flegg

Section Editor

PLOS Computational Biology

Jennifer Flegg

Section Editor

PLOS Computational Biology

Reviewer's Responses to Questions

**Comments to the Authors: 
Please note here if the review is uploaded as an attachment.**

Reviewer #1: Thank you for addressing my comments and critique, I appreciate the schematics.

Reviewer #2: All comments have been considered or addressed.

**Have the authors made all data and (if applicable) computational code underlying the findings in their manuscript fully available?**

Reviewer #1: Yes

Reviewer #2: Yes

PLOS authors have the option to publish the peer review history of their article (what does this mean?). If published, this will include your full peer review and any attached files.

Reviewer #1: No

Reviewer #2: No

---

## [Editor Report · Acceptance letter]

PCOMPBIOL-D-25-01479R1

Quantifying vector diversion effects in zoonotic systems: A modelling framework for arbovirus transmission between reservoir and dead-end hosts

Dear Dr Fairbanks,

I am pleased to inform you that your manuscript has been formally accepted for publication in PLOS Computational Biology. Your manuscript is now with our production department and you will be notified of the publication date in due course.

With kind regards,

Judit Kozma
